# Identifying domains of applicability of machine learning models for materials science

Christopher Sutton 1,7✉, Mario Boley 2,7✉, Luca M. Ghiringhelli 1✉, Matthias Rupp 1,3,6, Jilles Vreeken 4 & Matthias Scheffler1,5

Although machine learning (ML) models promise to substantially accelerate the discovery of novel materials, their performance is often still insufficient to draw reliable conclusions. Improved ML models are therefore actively researched, but their design is currently guided mainly by monitoring the average model test error. This can render different models indistinguishable although their performance differs substantially across materials, or it can make a model appear generally insufficient while it actually works well in specific sub-domains. Here, we present a method, based on subgroup discovery, for detecting domains of applicability (DA) of models within a materials class. The utility of this approach is demonstrated by analyzing three state-of-the-art ML models for predicting the formation energy of transparent conducting oxides. We find that, despite having a mutually indistinguishable and unsatisfactory average error, the models have DAs with distinctive features and notably improved performance.

[1] NOMAD Laboratory, Fritz Haber Institute of the Max Planck Society, Berlin, Germany. [2] Faculty of IT, Monash University, Clayton VIC 3800, Australia. [3] Citrine Informatics, Redwood City, CA 94063, USA. [4] CISPA Helmholtz Center for Information Security, Saarbrücken, Germany. [5] Physics Department, IRIS Adlershof Humboldt-Universität, Berlin, Germany. [6]Present address: Department of Computer and Information Science, University of Konstanz, Konstanz, Germany. [7]These authors contributed equally: Christopher Sutton, Mario Boley. ✉email: sutton@fhi-berlin.mpg.de; mario.boley@monash.edu; ghiringhelli@fhi-berlin.mpg.de

Given sufficient predictive accuracy, machine learning (ML) can accelerate the discovery of novel materials by allowing to rapidly screen compounds at orders of magnitude lower computational cost than first-principles electronic-structure approaches[1–7]. In practice, however, the accuracy of ML models is often insufficient to draw reliable conclusions about materials for specific applications[7]. Therefore, different ML representations for materials are actively developed to provide accurate predictions over diverse materials classes and properties[8–20]. A critical obstacle for this effort is that the complex choices involved in designing ML models are currently made based on the overly simplistic metric of the average model test error with respect to the entire materials class. This treatment of models as a black box that produces a single error statistic can render different models indistinguishable although their performance actually differs substantially across materials. Moreover, models may appear generally insufficient for certain screening tasks while they actually predict the target property accurately in specific subdomains. For example, for a large public ML challenge for predicting the formation energies of transparent conducting oxides (TCOs)[21], three approaches have a nearly indistinguishable performance: the competition winning model adapted from natural language processing (n-gram method)[21], smooth overlap of atomic positions (SOAP)[13,14], and the many-body tensor representation (MBTR)[12]. Importantly, as shown below, they all appear unsatisfactory for screening applications as they fail to reliably identify the ground state polymorph structure for many of the examined systems.

Here we present an informed diagnostic tool based on subgroup discovery (SGD)[22–24] that detects domains of applicability (DA) of ML models within a materials class. These domains are given as a combination of simple conditions on the unit-cell structure (e.g., on the lattice vectors, lattice angles, and bond distances) under which the model error is substantially lower than its global average in the complete materials class. Thus, in contrast to methods that provide uncertainty estimates for individual data points (such as probabilistic models or ensemble methods), the presented approach provides logical descriptions of contiguous regions with an overall low estimated uncertainty. These descriptions allow (a) to understand and subsequently address systematic shortcomings of the investigated ML model and (b) to focus sampling of candidate materials on regions of low expected model uncertainty. We demonstrate this procedure by analyzing the three state-of-the-art ML models for the above mentioned TCO challenge. Despite having a globally indistinguishable and unsatisfactory average error, the models have domains of applicability (DAs) with notably improved performance and distinctive features. That is, they all perform well for different characteristics of the unit cell. In terms of error improvement, the MBTR-based model stands out with a ca. twofold reduction in the average error and ca. 7.5-fold reduction in the fraction of errors above the required accuracy to identify the ground state polymorph (i.e., from 12.8 to 1.7%). Thus, we demonstrate that the MBTR-based model is in fact feasible for screening materials that lie within its DA while it is highly unreliable outside of it. This illustrates how the proposed method can be used to guide the development of ML representations through the identification of their systematic strengths and weaknesses. We expect this form of analysis to advance ML methods for materials as well as ML methods for science more broadly.

## Results

### Domain of applicability identification via subgroup discovery.

To formally introduce the method for DA identification, we recall some notions of ML for materials. In order to apply smooth function approximation techniques like ridge regression, the materials of interest are represented as vectors in a vector space $X$ according to some chosen representation. The more complex state-of-the-art representations evaluated in this work are defined further below. A first simple example is to use features $\phi_1, \ldots, \phi_n$ of the isolated atoms that constitute the material (e.g., $\phi_i(Z)$ may be the "electronegativity of the species with atomic number $Z$" (see Supplementary Table 4) and then to lift these to representation coordinates $x_i$ for compounds $(Z_j, \mu_j)_{j=1}^k$ defined as

$$x_i = \sum_{j=1}^{k} \mu_j \phi_i(Z_j) \qquad (1)$$

where $\mu_j$ corresponds to the mixture coefficient for atomic number $Z_j$. Moreover, let $y$ be a numeric material property according to which screening should be performed (in this work, we focus on formation energy, which is relevant for performing a ground state search).

A predictive ML model is then a function $f : X \to \mathbb{R}$ aiming to minimize the expected error (also called prediction risk)

$$e(f) = \int_{X \times \mathbb{R}} l(f(\boldsymbol{x}), y) dP(\boldsymbol{x}, y) \qquad (2)$$

measured by some non-negative loss function $l$ that quantifies the cost incurred by predicting the actual property value $y$ with $f(x)$. Examples for loss functions are the squared error ($l(y', y) = (y' - y)^2$), the absolute error ($l(y', y) = |y' - y|$), and, for non-zero properties, the relative error ($l(y', y) = |y' - y|/|y|$). Here $P$ denotes some fixed probability distribution that captures how candidate materials are assumed to be sampled from the materials class (this concept, while commonly assumed in ML, is an unnecessary restriction for high-throughput screening as we discuss in more detail below). Since the true prediction risk is impossible to compute directly without perfect knowledge of the investigated materials class, models are evaluated by the test error (or empirical risk)

$$\hat{e}(f) = \sum_{i=1}^{m} e_i(f)/m \qquad (3)$$

defined as the average of the individual errors (losses) $e_i(f) = l(f(x_i), y_i)$ on some test set of $m$ reference data points $(x_i, y_i)_{i=1}^m$. The samples in this test set are drawn independently and identically distributed according to $P$ and are also independent of the model—which means in practice that it is a random subset of all available reference data that has been withheld from the ML algorithm. In order to reduce the variance of this estimate, a common strategy is cross-validation, where this process is repeated multiple times based on partitioning the data into a number of non-overlapping "folds" and then to use each of these folds as test sets and the remaining data as a training set to fit the model.

This test error properly estimates the model performance globally over the whole representation space $X$ (weighted by the distribution $P$ used to generate the test points). This is an appropriate evaluation metric for selecting a model that is required to work well on average for arbitrary new input materials that are sampled according to the same distribution $P$. This is, however, not the condition of high-throughput screening. Here, rather than being presented with random inputs, we can decide which candidate materials to screen next. This observation leads to the central idea enabled by the DA analysis proposed in this work: if the employed model is particularly applicable in a specific subdomain of the materials class, and if that subdomain has a simple and interpretable shape that permits to generate new materials from it, then we can directly focus the screening there.

Such simply described DA can be identified by the descriptive data mining technique of subgroup discovery (SGD)[22–24]. This technique finds selectors in the form of logical conjunctions, i.e.,

Boolean functions ($\sigma$: $X \rightarrow$ {true, false}) of the form:

$$\sigma(x) \equiv \pi_1(x) \wedge \pi_2(x) \wedge \ldots \wedge \pi_p(x)$$

where "$\wedge$" denotes the "and" operation and each proposition $\pi_i$ is a simple inequality constraint on one of the coordinates, i.e., $\pi_i(x) \equiv x_j \leq v$ for some constant $v$. Thus, these selectors describe intersections of axis-parallel half-spaces resulting in simple convex regions ({$x \in X$: $\sigma(x) =$ true}) in $X$. This allows to systematically reason about the described subdomains (e.g., it is easy to determine their differences and overlap) and also to sample novel points from them. To specifically obtain regions where a given model has a decreased error, standard SGD algorithms[25,26] can be configured to yield a selector with maximum impact on the model error. The impact is defined as the product of selector coverage, i.e., the probability of the event $\sigma(x) =$ true, and the selector effect on the model error, i.e., the model error minus the model error given that the features satisfy the selector.

**An illustrative example**. Before describing the details of DA identification and its integration into the ML process, let us illustrate the concept and its utility via a synthetic example (see Fig. 1). We consider a simple two-dimensional representation consisting of independent features $x_1$ and $x_2$ that are each distributed according to a normal distribution with mean 0 and variance 2 ($N(0, 2)$) and a target property $y$ that is a third-degree polynomial in $x_1$ with an additive noise component that scales exponentially in $x_2$:

$$y \sim x_1^3 - x_1 + N(0, \exp(x_2/2)).$$

That is, the $y$ values are almost determined by the third-degree polynomial for low $x_2$ values but are almost completely random

for high $x_2$ values. Discovering applicable domains reveals how different models cope differently with this setting even if they have a comparable average error. To show this, let us examine the error distributions obtained from three different kernelized regression models of the form

$$f(\cdot) = \sum_{i=1}^{n} \nu_i k(x_i^F, \cdot)$$

with parameter vector $\boldsymbol{\nu}$ that are fitted around a training, or fitting ($F$), set $(x_i^F, y_i^F)_{i=1}^{n}$ with three different choices for the kernel function $k$. We observe:

- When using the linear (lin) kernel ($k(x, x') = \langle x, x' \rangle$), the resulting linear model is globally incapable to trace the variation of the third-order polynomial except for a small stripe on the $x_1$-axis where it can be approximated well by a linear function. Consequently, there is a very high error globally that is substantially reduced in the DA described by $\sigma_{\text{lin}}(x_1, x_2) \equiv -0.3 \leq x_1 \leq 0.3$.
- When using the Gaussian kernel $k(x, x') = \exp \| x - x' \|^2 / 2\epsilon^2$), the resulting radial basis function (rbf) model is able to represent the target property well locally unless (a) the noise component is too large and (b) the variation of the target property is too high relative to the number of training points. The second restriction is because the rbfs have non-negligible values only within a small region around the training examples. Consequently, the discovered DA is not only restricted in $x_2$-direction but also excludes high absolute $x_1$-values: $\sigma_{\text{rbf}} \equiv -3.3 \leq x_{11} \leq 3.1 \wedge x_2 \leq 0.1$.
- In contrast, when using the non-local third-degree polynomial (poly) kernel $k(x, x') = (\langle x, x' \rangle + 1)^3$, data sparsity

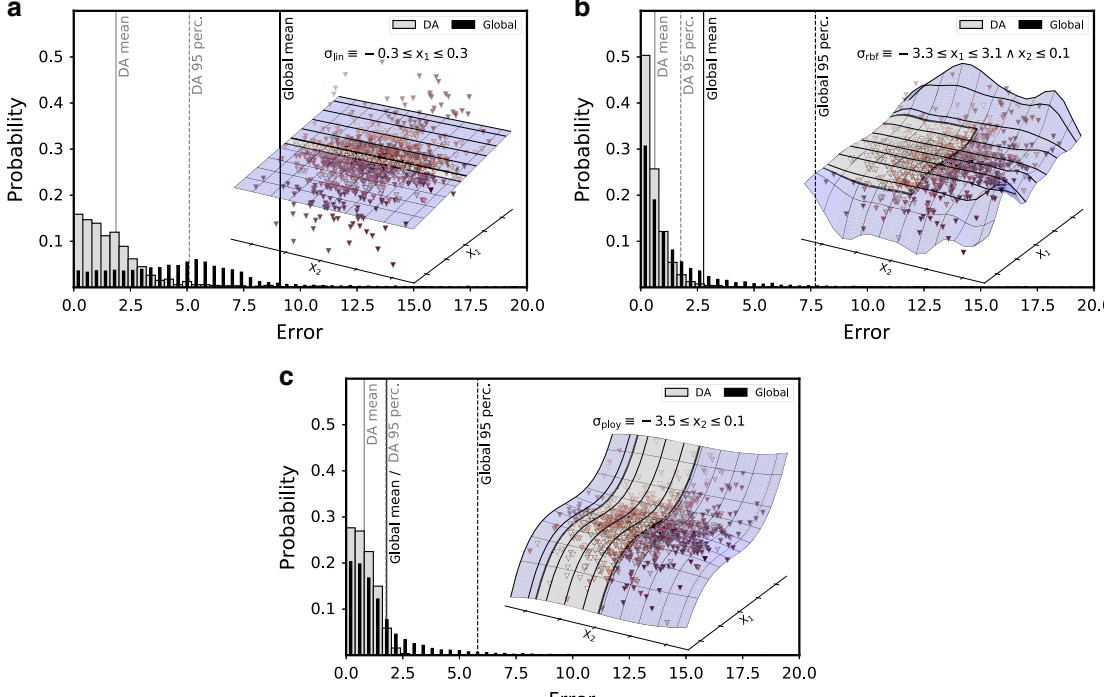

**Fig. 1 Domains of applicability of three 2d-models of a noisy third-degree polynomial.** Three different models, linear (top), radial basis function (rbf, center), and polynomial (poly, bottom), are shown approximating the same distribution of two independent features $x_1 \sim N(0, 2)$ and $x_2 \sim N(0, 2)$, and the target property $y \sim x_1^3 - x_1 + N(0, \exp(x_2/2))$, where $N(\mu, \epsilon^2)$ denotes a normal distribution with mean $\mu$ and standard deviation $\epsilon$. Test points are plotted in 3d plots against the prediction surface of the models (color corresponds to absolute error) where the DA is highlighted in gray. The distributions of individual errors for the DA (gray) and globally (black) are shown in the 2d plots of each panel with the mean error (solid) and the 95th percentile (95 perc./dashed) marked by vertical lines. Note that the global error distribution of the linear model has a considerably long tail, which is capped in the image.

                    

| Table 1 Features used for discovery of domain of applicability (DA) selectors. | | | |
|---|---|---|---|
| **Type** | **Label** | **Definition** | **Unit** |
| Unit cell | $a, b, c$ | Lattice-vector lengths sorted from largest ($a$) to smallest ($c$) | Å |
| | $\alpha$ | Angle between $b$ and $c$ | ° |
| | $\beta$ | Angle between $a$ and $c$ | ° |
| | $\gamma$ | Angle between $a$ and $b$ | ° |
| | $V/N$ | Volume of unit cell divided by number of atoms | Å³ |
| | $N$ | Number of atoms | – |
| Composition | %Al, %Ga, %In | Number of cations divided by total number of cations | % |
| Structural | $R_{\{Al,Ga,In\}-\{Al,Ga,In,O\}}$ | Average nearest-neighbor distance between Al, Ga, In, and O | Å |

does not prevent an accurate modeling of the target property along the $x_1$-axis. However, this non-locality is counter-productive along the $x_2$-axis where overfitting of the noise component has a global influence that results in higher prediction errors for the almost deterministic data points with low $x_2$-values. This is reflected in the identified DA $\sigma_{\text{poly}}(x_1, x_2) \equiv -3.5 \leq x_2 \leq 0.1$, which contains no restriction in $x_1$-direction, but excludes both high and low $x_2$-values. This highlights an important structural difference between the rbf and the polynomial model that is not reflected in their similar average errors.

**DA representation and objective function**. In the illustrative example above, all evaluated models share the same simple representation. However, in practice different models are typically fitted with different and more complicated representations. For instance, for the study on formation energies of transparent oxides below, we compare models based on the $n$-gram, SOAP, and MBTR representations. These representations use different descriptions of the local atomic geometry, leading to high-dimensional non-linear transforms of the material configurations (e.g., 1400, 681, and 472 dimensions for MBTR, SOAP, and $n$-gram representations). A DA described directly in terms of these complex representations cannot easily be mapped back to intuitive conditions on the unit cell of a given material. This not only hinders interpreting the DA but also to construct novel materials from it. Finally, using different representations to describe DAs of different models makes it impossible to assess their overlap and differences. Therefore, we define a single representation comprised of features that are specifically intended for the description of insightful subdomains. A first natural group of features pertains directly to the shape of the unit cell such as the sorted lattice vectors and angles, the number of atoms in the unit cell, and the unit-cell volume. In addition, when we are interested in a fixed compositional space, we can add features describing the composition (e.g., "percentage of Al cations") as well as structural features describing the bonding environments (e.g., "average nearest-neighbor distance between Al and O", which we define using the effective coordination number[27]). The description of DAs in these simple terms of the unit-cell structure and composition allows to easily interpret, compare, and sample from them (e.g., for focused screening). However, we note that the representation space inputted into subgroup discovery can be adapted for various purposes depending on the focus of the investigation. See Table 1 for a summary of all features used.

The DA optimization and validation can be performed as a by-product from the labels and ML predictions of the test set. However, just as for the ML-model fitting itself, we can only estimate these quantities based on empirical data. For that purpose, it is sensible to also split the test data into two parts: a DA identification set for optimizing the empirical impact and a

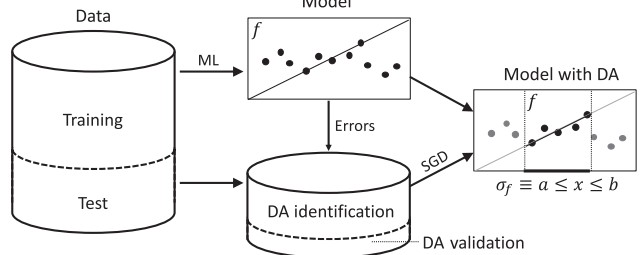

**Fig. 2 Workflow for domain of applicability (DA) identification and validation for an ML model.** The DA is described by a selector ($\sigma_f$) that is comprised of logical conjunctions of a representation space (here symbolized by a single dimension $x$ for simplicity but may be multidimensional). The selector is identified by applying subgroup discovery (SGD) to the individual ML-model errors for subset of test set (DA identification set). An unbiased estimate of the model performance within the DA is obtained on the remaining samples of the test set that were left out of the DA identification (DA validation set).

DA validation set for obtaining an unbiased performance estimate of the identified DA (see Fig. 2 for an illustration of the overall workflow). Technically, the data points withheld in the DA validation set mimic novel independent sample points that can be used to evaluate both the coverage of the DA, as well as, the reduction in model error. As an extension of this, one can also repeat the DA optimization/validation on several splits (cross-validation) to reduce the variance of the coverage and model error estimates and, moreover, to assess the stability of the DA selector elements.

For ease of notation we assume the DA identification set consists of the first $k$ points of the test set. We end up with the following objective function for the SGD algorithm:

$$\text{impact}(\sigma) = \underbrace{\left(\frac{s}{k}\right)}_{\text{coverage}} \underbrace{\left(\frac{1}{k}\sum_{i=1}^{k} l_i(f) - \frac{1}{s}\sum_{i \in I(\sigma)} l_i(f)\right)}_{\text{effect on test error}} \quad (4)$$

where $s$ denotes the number of points in the DA identification set selected by $\sigma$ and $I(\sigma) = \{i: 1 \leq i \leq k, \sigma(x_i) = \text{true}\}$ denotes the set of selected indices itself. Here, we focus on DA identification based on the relative error $l(y', y) = |y' - y|/|y|$, as it is less correlated with the target values than the absolute error. Thus, this choice promotes DAs that contain a representative distribution of target values and, by extension, more distinct and thus more characteristic DAs for the different models (see Supplementary Note 2 for a discussion of the DAs resulting from using the absolute error).

The effect term of the objective function ensures that the model is estimated to be more accurate in the described region than in

the global representation space. Thus, selectors with a large effect value describe domains of increased applicability as desired (see also Supplementary Note 4). In addition, promoting large, i.e., general, DAs through the coverage term is important as those have a higher chance to (a) contain data points of interest and (b) to have an accurate effect estimate, i.e., the empirical error reduction measured by the effect term is likely to generalize to other points in the DA that are not contained in the DA identification set. Thus, the coverage term has a similar role as a regularization term in common objective functions for model fitting. With the above objective function, we reduce the bi-criterial coverage/effect optimization problem to a uni-criterial impact optimization problem where both individual criteria are equally weighted and non-compensatory, i.e., due to the multiplicative combination, very low values of one criterion cannot be compensated by very high values in the other. The relative weight of both criteria can be re-calibrated by introducing a simple exponential weight parameter (see the Supplementary Methods section on Coverage/Effect Trade-off for a detailed discussion).

Optimizing the impact function over all conjunctive selectors that can be formed from a given set of base propositions is an NP-hard problem. This implies that there is no solver for it with worst-case polynomial time complexity (unless P = NP). However, there is a practically efficient branch-and-bound algorithm that turns out to be very fast in practice if the dimensionality of the DA representation is not too high—in particular, substantially faster than the model training process (see Methods and Supplementary Methods).

**Domains of applicability for TCO models.** Equipped with the DA concept, we can now examine the ML models for the prediction of stable alloys with potential application as transparent conducting oxides (TCOs). Materials that are both transparent to visible light and electrically conductive are important for a variety of technological devices such as photovoltaic cells, light-emitting diodes for flat-panel displays, transistors, sensors, touch screens, and lasers[28–38]. However, only a small number of TCOs have been realized because typically the properties that maximize transparency are detrimental to conductivity and vice versa. Because of their promise for technologically relevant applications, a public data-analytics competition was organized by the Novel Materials Discovery Center of Excellence (NOMAD[39]) and hosted by the on-line platform Kaggle using a dataset of 3000 $(Al_xGa_yIn_z)_2O_3$ sesquioxides, spanning six different space groups. The target property in this examination is the formation energy, which is a measure of the energetic stability of the specific elements in a local environment that is defined by the specific lattice structure.

Our aim is to demonstrate the ability of the proposed DA analysis to (i) differentiate the performance of models based on different representations of the local atomic information of each structure and (ii) to identify subdomains in which they can be used reliably for high-throughput screening. Specifically, we focus on the state-of-the-art representations of MBTR, SOAP, and the $n$-gram representation (all described in the Methods section). As an additional benchmark, we also perform DA identification for a simple representation containing just atomic properties averaged by the compositions (this corresponds to the simplistic choice of a representation given in Eq. (1); see Supplementary Table 4 for a list of atomic properties used in this representation). Since this representation is oblivious to configurational disorder (i.e., many distinct structures that are possible at a given composition), it is expected to perform poorly across all space groups and concentrations. Formally, there is no unique $y$-value associated with each $x$ but rather a distribution $P(y|x)$. Thus, even the optimal prediction at each composition of the test set (the median energy) to predict the test set energies results in a mean absolute error of 32.6 meV/cation, which is the highest accuracy that can be obtained using just composition-based properties. Therefore, it is a candidate for a representation that does not have any useful DA when compared with its full domain.

MBTR, SOAP, and $n$-gram all display a similar test error (using the absolute error as the loss function $l$ (see Eq. (3)); the resulting quantity we refer to as the mean absolute error, MAE) of 14.2, 14.1, and 14.7 meV/cation, respectively. This confirms previously reported virtually indistinguishable accuracies for MBTR and SOAP in the prediction of formation energies of alloys[40]. However, using the proposed method, key differences can be observed in the MAEs of their respective DAs (see Table 2 and Fig. 3 for a summary of all model performances). More specifically, the ML models built from MBTR, SOAP, and $n$-gram have an MAE (standard deviation) over the relevant DA validation sets of 7.6 (±1.5), 11.7 (±1.8), 10.2 (±0.9) meV/cation, respectively. All identified DAs for the models utilizing MBTR, SOAP, and $n$-gram have a large coverage (i.e., percent of samples within the DA) with an average (standard deviation) subpopulation contained within the DA validation set of 44% (±6%), 78% (±3%), and 52% (±5%), respectively.

In contrast, the atomic model is not only the worst model globally with a test error of 65.5 meV/cation, but, as anticipated, the DA error is virtually indistinguishable from the global model error (MAE = 60.2 meV/cation). This model performs worse than the MAE = 32.6 meV/cation that can be obtained by using the median energy at each composition of the test set to predict test set energies. Therefore, this result illustrates the case of a weak representation for which no DA with substantial error reduction can be identified.

Although the reduction of the mean error for the three state-of-the-art representations is notable, the difference between the whole materials space and the DAs is even more pronounced when comparing the tails of the error distributions using the 95th percentile. For the global models, the average 95th percentile across all relevant splits is reduced by a factor of 2.9, 1.4, and 1.6 for the DA compared with the global error for MBTR, SOAP, and $n$-gram.

**Table 2 Summary statistics for DAs for all investigated models.**

|  | Global (test set) | | | DA (validation set) | | | | | DA (identification set) | | |
|---|---|---|---|---|---|---|---|---|---|---|---|
|  | MAE | 95AE | R | cov | MAE | 95AE | R | cov | MAE | 95AE | R |
| MBTR | 14.2 | 54.1 | 0.83 | 44 (6) | 7.6 (1.5) | 18.8 (2.9) | 0.88 (0.03) | 44 (1) | 7.6 (0.3) | 20.7 (0.2) | 0.89 (0.01) |
| SOAP | 14.1 | 51.0 | 0.84 | 78 (3) | 11.7 (1.8) | 36.6 (10.8) | 0.85 (0.01) | 76 (1) | 11.9 (0.4) | 37.8 (2.0) | 0.85 (0.00) |
| $n$-gram | 14.7 | 51.1 | 0.83 | 52 (5) | 10.2 (0.9) | 32.6 (2.6) | 0.86 (0.02) | 54 (1) | 10.3 (0.2) | 35.5 (1.0) | 0.86 (0.00) |
| Atomic | 65.5 | 154.5 | 0.24 | 85 (1) | 60.2 (7.8) | 141.6 (28.5) | 0.25 (0.09) | 85 (0) | 63.3 (1.5) | 153.9 (5.5) | 0.25 (0.02) |

Coverage (cov), mean absolute error (MAE), 95th-percentile absolute error (95AE), and coefficient of determination based on absolute error (R) are all estimated via the mean value of the relevant DA validation sets and DA identification sets. Standard deviations are in parentheses. Global values are computed over whole test set. MAE and 95AE are in units of meV/cation, cov values are in percentages.

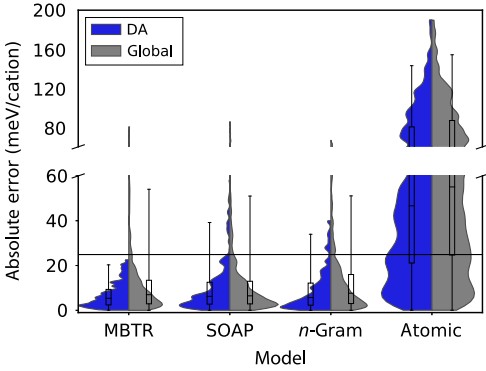

**Fig. 3 DA versus global ML-model performance.** Comparison of absolute error distributions for entire test set (global/gray) and for DA validation sets (DA/blue) using violin plots that extend to the 98th percentile. Boxplots inside violin plots indicate 25th, 50th, and 75th percentiles (boxes) of the absolute errors, as well as 5th and 95th percentiles (whiskers). Horizontal line indicates reference error level of half of the mean energy difference between the minimum energy and the second-to-minimum energy polymorph (mean over all considered concentrations).

To put these error values into context, we consider the reference value of 24.9 meV/cation corresponding to half of the mean energy difference between the minimum energy and the second-to-minimum energy polymorph for all concentrations. The fraction of data points with these errors from the MBTR model above this reference value is reduced by a factor of 7.5 from 12.8% in the entire test set to 1.7% (averaged over each relevant split) within the DA. A smaller reduction in the fraction of errors is observed for the SOAP model (13.3 to 9.0%) and $n$-gram model (16.2 to 10.8%). For the MBTR model, the 95th percentile of the DA errors (18.8 meV/cation) lies below the reference value.

Since the restriction of features values in the DAs generally affects the distribution of target values, the observed MAE improvements might simply be a function of reduced variation in the property values. This would mean that the DAs are not actually characteristic for the models as those reduced variations would be independent of them. However, comparing the coefficient of determination (R) of the models globally and within their DA reveals that this is not the case. The R values are increased from 0.83 to 0.88 (MBTR), 0.84 to 0.85 (SOAP), and 0.83 to 0.86 ($n$-gram). This means, while there is a reduction in target dispersion, there is a disproportionate reduction in MAE in the model-specific predictions. Note that, matching our interest in absolute error performance, we consider here the R-value defined as one minus the sum of absolute errors over dispersion measured as the sum of absolute deviations from the median[41].

The identified selectors are mostly stable, i.e., appearing in four out of six splits for MBTR and SOAP and five of six for $n$-gram. Interestingly, the variables that comprise the selectors of the DA are qualitatively different for each of these models. Selectors for MBTR include the number of atoms ($N$), the angle between the two longest lattice vectors in the unit cell ($\gamma$), and the average bond distance between aluminum and oxygen within the first coordination shell (that is defined by the effective coordination number), $R_{\text{Al-O}}$:

$$\sigma_{\text{MBTR}} \equiv N \geq 50 \text{ atoms} \land \gamma \leq 98.83° \land R_{\text{Al-O}} \leq 2.06 \text{ Å}.$$

For SOAP, selectors include features exclusively based on the unit-cell shape such as the ratio of the longest ($a$) and shortest ($c$) lattice vectors, and lattice-vector angles ($\beta$ and $\gamma$):

$$\sigma_{\text{SOAP}} \equiv \frac{a}{c} \leq 3.87 \land \gamma < 90.35° \land \beta \geq 88.68°$$

The selector of the $n$-gram model includes both features describing the unit-cell shape [medium lattice vector ($b$) and angle ($\gamma$)] and structural motifs [interatomic bond distances between Al–O ($R_{\text{Al-O}}$) and Ga–O ($R_{\text{Ga-O}}$) within the first coordination shell]:

$$\sigma_{\text{n-gram}} \equiv b \geq 5.59 \text{ Å} \land \gamma < 90.35° \land$$
$$R_{\text{Al-O}} \leq 2.06 \text{ Å} \land R_{\text{Ga-O}} \leq 2.07 \text{ Å}$$

It is worth noting that applying these DA selectors to the training set results in a similar reduction in error between the global and local populations and sample coverages (i.e., local population size) to what was observed for the test set: The training MAEs are reduced by factors of 1.87, 1.18, and 1.43 and the training DA coverages are 44%, 76%, and 54% for MBTR, SOAP, and $n$-gram models, respectively.

The qualitative differences observed in the DA selectors for these three models can be quantified by examining the overlapping samples in the DAs using the Jaccard similarity, which is the ratio of the number of overlapping samples over the total number of samples in both DAs. We find Jaccard similarities of 0.61 for $n$-gram vs. SOAP, 0.66 for $n$-gram vs. MBTR, 0.57 for SOAP vs. MBTR (computed over the whole test set). In other words, the discovered DA selectors are not only syntactically different, but, despite some overlap, they do indeed describe substantially different sub-populations of the investigated materials class.

We close this results section by an investigation of the effect of the individual DA selector elements of the SOAP-based model (details for MBTR and $n$-gram based models are provided in Supplementary Figs. 1 and 2). The inclusion of the attributes $\gamma < 90.35°$ and $\beta \geq 88.68°$ excludes 18.3% and 1.8% samples that have irregular unit cells based on the relatively large $\gamma$ and small $\beta$ values compared with the rest of the data points (see Fig. 4 for the distribution of the selected $a/c$, $\gamma$, and $\beta$ values for the SOAP-based model). The inclusion of the term $a/c \leq 3.87$, which describes 86.2% of the test set, is attributed to the fact that SOAP employs a real-space radial cut-off value $r_{\text{cut}} = 10$ Å in constructing the local atomic density for all samples (see above for a description of this representation). The algorithm threshold choice of $a/c \leq 3.87$ separates two modes of a relatively dense region of points (see Fig. 4 left panel); however, for structures with asymmetric unit cell, the spherical radius could lead to inaccurate depiction of the local atomic environment, therefore, we repeat the procedure for two additional $r_{\text{cut}}$ values of 20 and 30 Å. Compared with the selector identified for $r_{\text{cut}} = 10$ Å, a largely consistent selector is observed when the cut-off value is changed to a value of $r_{\text{cut}} = 20$ Å:

$$\sigma_{\text{SOAP},r_{\text{cut}}=20\text{Å}} \equiv \frac{a}{c} \leq 3.89 \land \gamma \leq 90.35°$$

However, increasing $r_{\text{cut}}$ to a value of 30 Å—which exceeds the largest unit-cell vector length ($a$) of ca. 24 Å in the structures contained within this dataset—results in the selector:

$$\sigma_{\text{SOAP},r_{\text{cut}}=30\text{Å}} \equiv c \geq 4.05 \text{ Å} \land \gamma \leq 90.35°$$

The absence of the $a/c$ term for the SOAP representation utilizing a $r_{\text{cut}} = 30$ Å indicates that the choice of a cut-off value less than the length of the unit cell directly impacts the model performance for the larger unit cells within this dataset, and thus, directly affects the selector chosen by SGD.

## Discussion
The presented approach identified DAs for each investigated model with notably improved predictive accuracy and a large

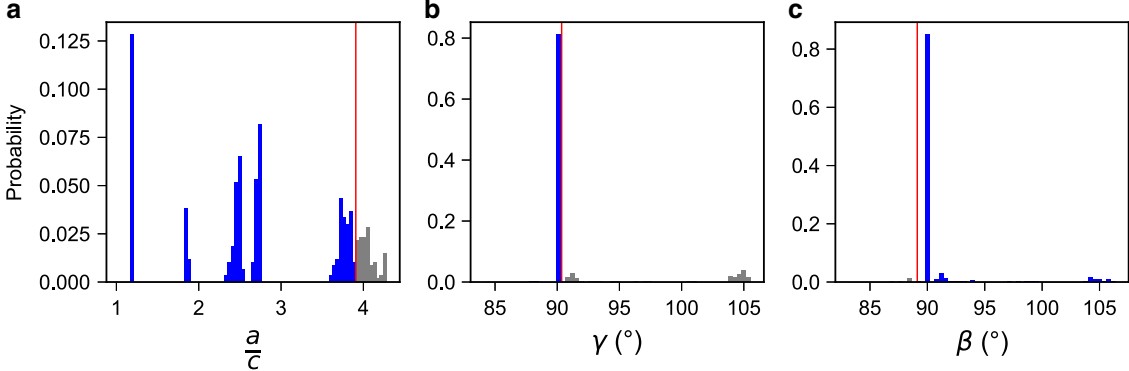

**Fig. 4 Effect of individual DA selector conditions for SOAP-based model.** The distributions of the three features referenced in the selector $\sigma_{SOAP} \equiv a/c \leq 3.87 \wedge \gamma < 90.35° \wedge \beta \geq 88.68°$, i.e., $a/c$ (**a**), $\gamma$ (**b**), and $\beta$ (**c**), are shown with subpopulation selected by condition in blue, subpopulation deselected in gray, and threshold by red line.

coverage of the underlying materials class (44–78%). In particular, the MBTR model displays a subdomain with a 95th-percentile error that is about a factor of 3 smaller than its global 95th-percentile error. Besides these quantitative assessments, the discovered DAs enable a qualitative comparison of the three investigated representations of materials by analyzing their defining logical formulas. These show notable differences that can be attributed to variation in the physics captured by the models. For example, the appearance of the number of atoms in the selector for MBTR indicates preferred fitting of specific unit-cell sizes due to the dependence on an unnormalized histogram in the representation. For SOAP, the selectors include features exclusively based on the unit-cell shape, which is attributed to the choice of a cut-off radius in the construction of the local atomic environment. In order to be applicable to a wider domain, improved versions of these representations need to address those systematic shortcomings—a conclusion which is illustrative of how the method of DA identification can guide the improvement of material representations and ML methods in general.

A further potential application of the proposed approach is to form ensembles of different local models, each of which is only contributing to the overall predictions for data points inside its DA. The general approach of forming ensembles of local models is an emergent idea to cope with heterogeneous materials classes[42,43]. So far these efforts have focused on a priori partitioning of the training set into sub-populations identified by automated clustering methods or prior knowledge, followed by fitting local models using the same regression technique for all subpopulations. In contrast, the DA approach can be used to incorporate the relative model advantages and disadvantages into the partitioning of the materials class.

In this context, it is an intuitive expectation that an improved model can be obtained by fitting only training data from within the discovered DA. However, this is not true in general: points outside of the DA, while having a higher error on average, can still contribute positively to the prediction inside the DA. For instance, refitting to a training set trimmed according to the DA selectors of the three model types investigated here leads to a change in test MAE of −1.5 (MBTR), −1.0 (SOAP), and +0.1 (n-gram) meV/cation. That is, we see an improvement for the MBTR and SOAP models but a slight decline in model performance for the n-gram model. Note that, only the DA validation set can be used to obtain an unbiased error estimate of a refitted model because it contains the only data that is independent of the overall fitting and refitting process. The statistical considerations related to model refitting are an interesting subject for future investigations and a better understanding could lead to an iterative

refitting scheme where DAs are refined until convergence. Such a scheme could also contain an active learning component where additional data points are sampled from within the identified subdomains.

## Methods

**MBTR**. The MBTR representation space $X$ can vary depending on the employed many-body order (e.g., interatomic distances for a two-body model, and/or angles for a two- and/or three-body model, and/or torsions for up to four-body models[12]). The results reported herein are calculated using a representation consisting of broadened histograms of element counts (one-body terms) and pairwise inverse interatomic distances (two-body terms), one for each unique pair of elements in the structure (i.e., for this dataset: Al–Al, Al–Ga, Al–In, Al–O, Ga–Ga, Ga–In, Ga–O, In–In, In–O, and O–O). These are generated according to:

$$g_i(1/r) = \frac{1}{\sqrt{2\pi\epsilon_{atom}^2}} \sum_j \exp\left(-\frac{(1/r - 1/|r_i - r_j|)^2}{2\epsilon_{atom}^2}\right) w_{MBTR}(i,j)$$

where a normal distribution function is centered at each inverse distance between pairs of atoms $(1/r_{ij})$ to ensure smoothness of the representation. The function $w_{MBTR}(i,j)$ dampens contributions from atoms separated by large distances and is defined as $w_{MBTR}(i,j) = (1/r_{ij})^2$. The MBTR representation was generated using `QMMLpack`[12,44].

**SOAP**. The SOAP representation space is constructed by transforming pairwise atomic distances as overlapping densities of neighboring atoms and expanding the resulting density in terms of radial and spherical harmonics basis functions. The local density is modeled through a sum of gaussian distributions on each of the atomic neighbors $j$ of atom $i$:

$$\rho_i(r) = \sum_j \exp\left(-\frac{(r - r_{ij})^2}{2\epsilon_b^2}\right) w_{SOAP}(r)$$

where $j$ ranges over neighbors within a specific cut-off radius ($r_{cut}$) relative to $i$, where the cut-off function $w_{SOAP}$ is defined as:

$$w_{SOAP}(r) = \begin{cases} 1 & \text{, for } r \leq r_{cut} - d \\ \left(\cos\left(\pi \frac{r - r_{cut} + d}{d}\right) + 1\right)/2 & \text{, for } r_{cut} - d < r \leq r_{cut} \\ 0 & \text{, otherwise} \end{cases}$$

The density $\rho_i(r)$ is then expanded in terms of spherical harmonics $Y_{k,m}(r/|r|)$ and orthogonal radial functions $g_n(|r|)$:

$$\rho_i(r) = \sum_{n,k,m} c_{n,k,m} g_n(|r|) Y_{k,m}\left(\frac{r}{|r|}\right).$$

The number of coefficients $c_{n,k,m}$ is given by the choice of basis set expansion values. Rotationally invariant features are then computed from the coefficients of the expansion and averaged to create a single per-structure representation, forming the input space $X$. A real-space radial cutoff of $r_{cut} = 10$ Å and $\epsilon_b = 0.5$ Å are used in this work. The SOAP representation was computed with the `QUIPPY` package available at https://libatoms.github.io/QUIP/index.html.

**n-gram**. The n-gram features are generated using a histogram of contiguous sequences of nodes (i.e., atoms) that are connected by edges (i.e., bonds) in a crystalline graph representation. An edge between nodes in the crystalline graph

occurs if the interatomic distance in the 3D crystal is less than a pre-specified cut-off distance ($r_{cut}$) that is proportional to the sum of the ionic radii of the two species. The number of edges of a given node $i$ corresponds to its coordination environment ($CN_i$):

$$CN_i = \sum_j r_{i,j} \mathbf{1}(r_{i,j} < r_{cut}).$$

The parameter $r_{cut}$ was taken to be lattice dependent as used in ref. [21]. Here, only the cation coordination environment is considered, which is defined entirely by the number of oxygen atoms in the first coordination shell. The $n$-gram representation utilizes contiguous sequences of up to four nodes (see ref. [21] for a detailed description of this approach). An implementation of the $n$-gram model used here is available at https://analytics-toolkit.nomad-coe.eu/home/.

**Machine learning models**. All representations were combined with kernel ridge regression using the rbf kernel as implemented in `scikit-learn`, version 0.21.3 (https://pypi.org/project/scikit-learn/0.23.1/). That is, ML models $f_\nu(x) = \sum_{i=1}^n \nu_i \exp(-\| x^F - x_i^F \|^2/(2\epsilon^2))$ with parameter vector $\nu$ are found by minimizing the objective

$$\sum_{i=1}^n (f_\nu(x_i^F) - y_i^F)^2 + \lambda \nu^T K \nu$$

using a training set $(x_i^F, y_i^F)_{i=1}^n$ of $n = 2400$ points. Here, $K$ refers to the $n \times n$ kernel matrix with entries corresponding to the application of the rbf kernel to all pairs of training points, i.e., $K_{ij} = \exp(-\| x_i^F - x_j^F \|^2/(2\epsilon^2))$ for $1 \le i, j \le n$. The values for the two hyperparameters $\epsilon$ and $\lambda$ are determined through a grid search within an inner loop of fivefold cross-validation where, in each iteration, the training set is split into a parameter tuning and a parameter validation set.

**Domain of applicability analysis**. Each single run of DA identification and evaluation was performed by applying SGD using 500 random data points of the TCO test set (DA identification set) and then determining DA performance on the remaining 100 data points of the test set (DA validation set). For each ML model, a single DA selector was then determined by (i) performing six runs using sixfold cross-validation (i.e., the test set was randomly partitioned into 6 non-overlapping folds of 100 data points, and each fold was used as DA validation set once) and (ii) choosing the selector that resulted from a majority of runs. For step (ii), selectors were considered equivalent if they only differed in threshold values in some inequality conditions and these differences did not result in a different selection of data points on the whole test set. The selectors were identified using a DA identification set, but the DA performance was then assessed as the mean value of the performance measures on the relevant DA validation sets. For an increased robustness of results, the above process was carried out three times for each model and the median performance over those repetitions was reported. The DA majority DA selectors themselves did not vary across those repetitions.

**Subgroup discovery**. The subgroup discovery target variable for DA identification was the relative (main text results) and absolute (results in Supplementary Information) model residuals, respectively. The basic propositions for the potential subgroup selectors were formed by first applying the feature map from Table 1 and then finding thresholds for inequality constraints via 5-means clustering[45]. Conjunctive selectors were optimized with respect to the impact objective function as given in Eq. (4) using branch-and-bound search with equivalence-aware non-redundant branching[46] and tight (selection-unaware) bounding[25,26]. That is, selectors are found using (1) a branching operator that generates exactly one selector for any set of selectors describing the same subset of data points, and (2) a bounding function that provides an upper bound to the best attainable specialization of a given selector by assuming that all subsets of its extension can be exactly described by some refined selector. All computations were run using the subgroup discovery implementation in `realKD 0.7.2`. More details on selector optimization and a refined objective function are discussed in the Supplementary Information.

## Data availability

All datasets involved in this research are available via the first author's GitHub account at https://github.com/csutton7/ML_domain_of_applicability. The dataset used in this analysis was originally produced for the NOMAD 2018 Kaggle competition (https://www.kaggle.com/c/nomad2018-predict-transparent-conductors). The output of all materials computations are available in the NOMAD Repository (https://doi.org/10.17172/NOMAD/2019.06.14-1). The curated competition dataset including the test data is also available at https://github.com/csutton7/nomad_2018_kaggle_dataset. A web-app allowing to reproduce the results of the Kaggle competition can be found at https://analytics-toolkit.nomad-coe.eu/tutorial-kaggle-competition.

## Code availability

The SGD implementation is publicly available through the open source Java library `realKD 0.7.2` available at https://bitbucket.org/realKD/realkd/. Python scripts for all computations involved in this research are available via the first authors GitHub account at https://github.com/csutton7/ML_domain_of_applicability. All are provided under the MIT open source software license. See http://analytics-toolkit.nomad-coe.eu/ML_domain_of_applicability for an interactive tutorial that allows to reproduce the results of the DA analysis.

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

## Acknowledgements

This work received funding from the European Union's Horizon 2020 Research and Innovation Programe (grant agreement No. 676580), the NOMAD laboratory CoE, and ERC:TEC1P (No. 740233). C.S. gratefully acknowledges funding by the Alexander von Humboldt Foundation.

## Author contributions

M.B. conceived the idea. C.S. and M.B. designed the initial method and research, analyzed the initial results together with L.M.G., M.R., J.V., and M.S., and wrote the initial paper. M.B., C.S, L.M.G., and M.S. designed revised versions of the method and the research. All authors reviewed the paper and contributed to its final revision.

## Competing interests

The authors declare no competing interests.
