## [Peer Review File · Nature Communications]

Reviewers' comments:

Reviewer #1 (Remarks to the Author):

The manuscript describes a methodology to improve machine learning predictions of materials properties. The idea is to identify regions in feature space where the predictions provided by the machine are not reliable, and discard those predictions. The authors test this approach on three entries of a recent Kaggle competition and demonstrate that indeed prediction errors go down considerably.

In my opinion, the results are not unexpected: if one has a prediction of the uncertainty of a result on a large data set, and exclude those where the uncertainties are large, the variance must go down, provided the uncertainty estimation was reasonable. I also think there are plenty of other established ways to estimate uncertainty in machine learning predictions, for example query by committee or other variance estimators. One would expect these to work equally well in this context.

I am also not convinced by the usefulness of this approach (whether it is based on subgroup discovery or by uncertainty estimates) other than identifying shortcomings of the representations and/or the machine learning approach - which should really be fixed rather than dodged. Often the reasoning for a high-throughput machine learning approach like the one simulated in the Kaggle competition is to find a needle in a haystack. If we discard a good portion of the haystack, we have just lowered the chances of finding the needle. Even worse, one could expect the out-of-ordinary elements to be the interesting ones from the scientific point of view.

I find it is a valid argument that this approach can be used to identify weaknesses in descriptors and improve them - but I think this may better belong to a more specialised journal. I am not sure yet another paper chasing lower mean absolute energy errors is of broad enough interest for the audience of Nature Communications.

Reviewer #2 (Remarks to the Author):

The paper presents an application of subgroup discovery to several machine learned models which predict crystal formation energies. For those datasets, the authors shows significant reductions in MAE when the predictions are constrained to sizable fractions of the original data. This work should be of interest to the field because it demonstrates the application of a technique that I have not seen anyone use in predictive ML for materials and could be relevant to almost any such application.

Broadly, I have no substantial concerns with the paper. I have a few suggestions for presentation, but these are all minor/easy to fix (see below).

The statistical analysis and machine learning process is appropriate and relatively standard. Because of the large number of details inherent in a study like this, I hope the authors will be releasing code alongside the publication.

The only unusual technical point is the mismatch in loss functions.

- * KRR uses an L2 norm
- * The analysis focuses on MAE (an L1 norm)
- * The SGD uses a relative error loss function

Any of these is fine, but it's unusual to see this. Perhaps the authors could explain why they made this

choice?

This comes up specifically for “the optimal prediction at each composition of the test set (the median energy)”. This is true that median is L1 optimal, but it’s not L2 optimal (which is what the training algorithm is optimizing)

Minor points

* “also to sample novel points from them” -- This statement is not true. Just because you can describe a sub-domain does not mean it is easy to sample (unless you are sampling from an enumerated set). Consider if the DA has said “PBE band gap energy > x” — finding such a material may be hard.

* In the section on Jaccard similarity, there is a reference to Figure S1 but Figure S1 does not show what is referenced here (I think you left a figure out of supplemental)

* Table S2 contains the key results from the paper and shouldn’t be relegated to supplemental! I think they could be relatively easily incorporated into Figure 3 if you are trying to avoid another main text figure/table.

* in the deeper dive in the SOAP DA model, you should probably reference Figures S1 and S2 for the other models.

* Captions for Figures S1 and S2 should describe what the histograms are. I think I know, but don’t make me guess.

Patrick Riley

Google Accelerated Science

Reviewer #3 (Remarks to the Author):

The manuscript at hand presents a novel approach to discover the applicability domain of trained machine learning (ML) models on the materials data sets. This approach is based on the previously presented method of sub-group discovery, which is equivalent to unsupervised ML techniques. The approach is tested on a dummy synthetic data and a small materials data set of formation energies of transparent conducting oxides. The final goal is to identify the domain of applicability (DA) of three kernel-based ML models. As it’s clearly stated, the challenge of DA in any ML research is timely and of interest. Thus, I believe the approach is relevant and arguably new. The text is also well written and it’s easy to understand. It conveys a generally acceptable ML workflow for training and validation. However, the discussions in the paper are mainly focused on the application of the DA approach as a fair metric to compare models and feature representations. I believe a general approach to DA is way more important than its application in comparing methods on its basis. There are a few recent research papers that try to address this issue from different aspects:

- Kailkhura et. al., Reliable and Explainable Machine Learning Methods for Accelerated Material Discovery, Arxiv 2019

- Haghghatlari et. al., Thinking Globally, Acting Locally: On the Issue of Training Set Imbalance and the Case for Local Machine Learning Models in Chemistry, ChemRxiv 2019

Since it’s difficult to provide benchmark data for unsupervised techniques, these references can be used to evaluate the generalizability of the proposed approach. Considering the novelty of the work, I recommend the following major concerns and comments be addressed before any potential publication:

1) It seems that each model is trained only once (on one of the training folds) and the SGD-DA approach will be applied to the errors of the model and splits of the test set. It is not clear that the prediction of the test set by ML model is required to validate the DA or not? If yes please also comment on the computational cost for each of the three feature representation methods.

2) The follow-up comment is on the optimization algorithm in the SGD method. I understand the objective function is explained here, but it seems that the core of the work (optimization algorithm) has been left out based upon the previously developed, SGD, technique. However, authors spend a generous amount of space to explain feature representations that are developed by other researchers.

3) One of the main concerns is regarding the limited number of features that are used as DA selectors. Although authors justify the neutrality and interpretability of the features, there is a place of concern for the correlation between these features and actual features that have been used for the training. This can be discussed from three different angles:

a) The choice of selectors may be biased to particular domain knowledge and perform in favor of one of the descriptors. This may be a valid point from a chemist point of view, but I believe it's not really fair for the sake of comparison. One can choose selectors that are more correlated with their choice of feature representation.

b) Although the actual features may not be chemically interpretable, they are the same features that one needs to calculate before using the model to predict energies. Thus, it sounds Ok to identify DA based on the same features.

c) The number of descriptors in each of the methods are not same. MBTR provides at least two times more descriptors than the other two (1400 vs 600). This is way more amount of information extracted to be interpreted in a small subgroup and against a limited number of selectors.

4) The other main concern is about the interpretation of the results. As mentioned in the text, the coverage of the MBTR is almost half of the SOAP model/representation (%44 vs %76). Thus, the MBTR model is applicable in a subspace of the data, which is half of the size of coverage by SOAP. Thus the DA of the MBTR is too restricted. I think a higher coverage of the training data is more desirable for an ML model with acceptable performance. I suspect if we shrink the subspace of the SOAP model elaborately (e.g., from %76 to %50), the number of outliers and thus the subgroup error will also shrink.

In addition, there are a number of minor issues that should be addressed before the paper is accepted:

1) on page 6, line 2: impact needs to be italic?

2) on page 9: Figure2, in the caption: the standard deviation based on the mathematical expression is ϵ not s

3) Please comment on the accessibility of the codes? or any GitHub repository for SGD/DA method?

We have copied the reviewers' reports below and included our replies, noting the changes made to the manuscript and SI.

Reviewer 1:

The manuscript describes a methodology to improve machine learning predictions of materials properties. The idea is to identify regions in feature space where the predictions provided by the machine are not reliable, and discard those predictions. The authors test this approach on three entries of a recent Kaggle competition and demonstrate that indeed prediction errors go down considerably.

R1.1 We would like to point out that the above is an important aspect of our work but only a partial summary of the scope of the DA approach. Crucially, it also enables an analysis of the factors that lead to the observed error reduction because the domains of applicability (DAs, the "identified regions") are described via logical conditions on simple descriptors mostly related to the unit cell structure. This allows for a both a quantitative and qualitative understanding of the "uncertainty landscape"; see also R1.3 for a quotation of the change in the introduction of the manuscript, where we have further clarified this.

In my opinion, the results are not unexpected: if one has a prediction of the uncertainty of a result on a large data set, and exclude those where the uncertainties are large, the variance must go down, provided the uncertainty estimation was reasonable.

R1.2 The reviewer brings up a good point that we have now clarified. A smaller error is to be expected to some degree as long as smaller DAs have a smaller variance (or more generally *dispersion*) of the target variable. However, as we show on Pg. 8 of the revised manuscript, using the R -values (see definition below) of the identified DAs, in all cases the reduction of error exceeds the reduction of dispersion:

"The R -values are increased from 0.83 to 0.88 (MBTR), 0.84 to 0.85 (SOAP), and 0.83 to 0.86 (n-gram). This means, while there is a reduction in target dispersion, there is a disproportionate reduction in MAE in the model-specific predictions."

Here, based on our interest in the MAE, we define R as the least-absolute-deviation (LAD) coefficient of determination, i.e., as 1 minus the ratio of absolute model errors to mean absolute deviations from the median.

I also think there are plenty of other established ways to estimate uncertainty in machine learning predictions, for example query by committee or other variance estimators. One would expect these to work equally well in this context.

R1.3 We agree with the reviewer that the literature is rich with variance-estimator-based methods for estimating uncertainties in machine learning predictions. However, variance estimators only provide estimates of the model uncertainty for single data points (feature vectors) without a descriptive summary of the “uncertainty landscape”. In contrast, the SGD-based approach provides a quantitative and qualitative description through the selector formed from logical conditions on the unit cell shape (e.g., on the lattice vectors, lattice angles, and bond distances). This is a significant difference and benefit of the SGD-based approach over conventional methods for uncertainty estimation, which we have emphasized more clearly on Pg. 1 of the revised manuscript:

“Thus, in contrast to methods that provide uncertainty estimates for individual data points (such as probabilistic models or ensemble methods), the presented approach provides logical descriptions of contiguous regions with an overall low estimated uncertainty. It is these descriptions that allow a) to understand and subsequently address systematic shortcomings of the investigated machine learning model and b) to focus sampling of candidate materials to regions of low expected model uncertainty.”

Often the reasoning for a high-throughput machine learning approach like the one simulated in the Kaggle competition is to find a needle in a haystack. If we discard a good portion of the haystack, we have just lowered the chances of finding the needle. Even worse, one could expect the out-of-ordinary elements to be the interesting ones from the scientific point of view.

R1.4 We feel that the reasoning of the reviewer is misleading because our message is that the “needle” can only be found when the description is sufficiently reliable. Identifying the DA of a set of candidate models will in fact increase the probability to *reliably* discover the “needle”, by reducing the risk of finding *false positives* in areas of the input space where the model is less accurate. Furthermore, excluding the domain outside the DA of a particular model as search area for the “needle” does not imply that such an area is neglected. Clearly a different model is needed whose DA covers at least part of the discarded region or an ensemble of ML models, which we discuss this possibility now on Pg. 9 as potential direction for future work (this is also included in our reply R3.3 to Referee 3):

“A further potential application of the proposed approach is to form ensembles of different local models, each of which is only contributing to the overall predictions for data points inside its DA. The general approach of forming ensembles of local models is an emergent idea to cope with heterogeneous material classes [Kailkhura et al., 2019, Haghightlari et. al., 2019]. So far these efforts have focused on a priori¹ partitioning of the training set into sub-populations identified by automated clustering methods or prior knowledge, followed by fitting local models using the same regression technique for all sub-populations. In contrast, the DA approach can be used to incorporate the relative model advantages and disadvantages into the partitioning of the materials class.”

¹That is, the sub-populations are found solely based on the feature and property values and are not defined based on the relative performance difference between different representations and regressors.

I find it is a valid argument that this approach can be used to identify weaknesses in descriptors and improve them—but I think this may better belong to a more specialised journal. I am not sure yet another paper chasing lower mean absolute energy errors is of broad enough interest for the audience of Nature Communications.

R1.5 We thank the reviewer for supporting our claim that the proposed method can be used to identify and improve general weaknesses in the model descriptors/representations. Thus, it should be fair to say that the proposed DA approach is not “yet another method chasing a lower MAE”. Instead, it is a robust and new tool for assessing where and why the ML model is more reliable. A crucial benefit of the SGD-based approach is the *description* of the DA. This is because the description offers the possibility to predict whether new data points will be reliably evaluated using the predictions from a given ML model without even evaluating the model descriptor/representation. We believe that assessing and analyzing the reliability of a given model is a useful tool for a very broad audience of ML practitioners. Our work also increases the awareness that the ML model is not trustworthy everywhere and outside of a given models DA, another description should be sought.

Reviewer 2:

The paper presents an application of subgroup discovery to several machine learned models which predict crystal formation energies. For those datasets, the authors shows significant reductions in MAE when the predictions are constrained to sizable fractions of the original data. This work should be of interest to the field because it demonstrates the application of a technique that I have not seen anyone use in predictive ML for materials and could be relevant to almost any such application.

R2.1 We thank the reviewer for their kind comments and support of the publication of this manuscript.

Broadly, I have no substantial concerns with the paper. I have a few suggestions for presentation, but these are all minor/easy to fix (see below). The statistical analysis and machine learning process is appropriate and relatively standard. Because of the large number of details inherent in a study like this, I hope the authors will be releasing code alongside the publication.

R2.2 All data and scripts involved in producing the results presented herein will be made available on github. An implementation of the SGD algorithm is publicly available through the realKD Java library at <https://bitbucket.org/realkD/realkd/> and a notebook illustrating the DA-identification procedure will be available at <https://analytics-toolkit.nomad-coe.eu>. This is now mentioned in the “Data and code availability” section of the main text.

The only unusual technical point is the mismatch in loss functions.

- **KRR uses an L2 norm**
- **The analysis focuses on MAE (an L1 norm)**
- **The SGD uses a relative error loss function**

Any of these is fine, but it’s unusual to see this. Perhaps the authors could explain why they made this choice? This comes up specifically for “the optimal prediction at each composition of the test set (the median energy)”. This is true that median is L1 optimal, but it’s not L2 optimal (which is what the training algorithm is optimizing).

R2.3 We thank the reviewer for their knowledgeable remark. Although regression techniques like Kernel Ridge Regression use the squared error for reasons of statistical and computational efficiency, the absolute error is the more descriptive measure of model performance because it quantifies the expected deviation of a prediction. Thus, it would indeed be a natural choice to directly optimize the effect on the absolute error for DA identification, and we now provide a detailed investigation of the DAs resulting from this choice in Section V of the SI. We observe that, indeed, the resulting error reductions are slightly higher, but also that the DAs have a less representative distribution of the prediction target and that there is more overlap between the DAs.

For the main text, the focus on the relative error was chosen because, in contrast to the absolute error, it is guaranteed to be uncorrelated with the target values. Thus, the choice of the relative error as the target promotes DAs that contain a representative distribution of target values and, by extension, more distinct and thus more characteristic DAs for the different models.

Motivated by the Reviewer 3’s remarks, we have discussed these considerations in the main text on Pg. 3:

“In contrast to the absolute error, the relative error is guaranteed to be uncorrelated with the target values. Thus, this choice promotes DAs that contain a representative distribution of target values and, by extension, more distinct and thus more characteristic DAs for the different models (see SI Section V for a discussion and an investigation of the DAs resulting from using the absolute error).”

Minor points: “also to sample novel points from them” – This statement is not true. Just because you can describe a sub-domain does not mean it is easy to sample (unless you are sampling from an enumerated set). Consider if the DA has said “PBE band gap energy > x” – finding such a material may be hard.

R2.4 This is a very good point. Using Computed PBE bandgap energies to identify new novel materials would indeed be problematic because of the computational cost of generating a given value. We now only use features that are generated entirely from unit cell and composition. Efficient sampling of novel structures from the DA can be performed using these features with a simple rejection sampling layer combined with a standard sampling approach. The specific values for each of the features in the selector are defined within the context of the distribution of the dataset. Sampling of novel points using the DA only make sense with if all new data are generated according to a distribution that is consistent with the initial model training/test set distribution. As a result of this change, the DA for the atomic property model, which previously was defined using bandgap energy, has become worse.

In the section on Jaccard similarity, there is a reference to Figure S1 but Figure S1 does not show what is referenced here (I think you left a figure out of supplemental)

R2.5 We thank the reviewer for finding this dead reference. This figure has been removed as it did not provide additional value over the numerical results provided in the main text.

Table S2 contains the key results from the paper and shouldn't be relegated to supplemental! I think they could be relatively easily incorporated into Figure 3 if you are trying to avoid another main text figure/table.

R2.6 These data are now provided in Table II of the main text. Note that the changed results for the atomic properties model are due to the revision of the available DA features (see R2.4). There were also some minor changes due re-partitioning the test set. Although the stability of the identified DAs was slightly reduced (for MBTR and SOAP the same selector identified in 4 of 6 splits, for n-gram is 5/6), coverage and error were only changed by insignificant amounts (e.g., for the error: -0.01 eV/cation for MBTR, +0.46 eV/cation for SOAP, -0.2 eV/cation for n-gram).

In the deeper dive in the SOAP DA model, you should probably reference Figures S1 and S2 for the other models.

R2.7 We have now clarified in the main text that these figures are provided in the SI on Pg. 8 of the main text: *"both MBTR and n-gram based models are provided in Figs. S1 and S2"*.

Captions for Figures S1 and S2 should describe what the histograms are. I think I know, but don't make me guess.

R2.8 Thanks for pointing this out. The caption for Figures S1 and S2 have now been updated.

All together, we are very grateful to the Reviewer for reading our paper so carefully and identifying several unclear aspects in the original manuscript, which has now been improved.

Reviewer 3:

The manuscript at hand presents a novel approach to discover the applicability domain of trained machine learning (ML) models on the materials data sets. This approach is based on the previously presented method of sub-group discovery, which is equivalent to unsupervised ML techniques.

R3.1 We would like to point out that the last statement is incorrect. Subgroup discovery is typically referred to as a *supervised* descriptive rule induction technique (see, e.g., Novak et al. "Supervised descriptive rule discovery: A unifying survey of contrast set, emerging pattern and subgroup mining." JMLR 10. Feb (2009): 377-403.). While the exact meaning of the terms "supervised" and "unsupervised" can vary in different application contexts, we would like to clarify that the approach uses feature variables x to partially characterize a target variable l , which is in our case the prediction loss incurred by an ML model when making a prediction for input x . In contrast, in unsupervised learning, and specifically in clustering, one typically does not have the differentiation between feature and target, thus the goal is to summarize the (empirical) distribution of a set of variables, e.g., by providing cluster representatives.

The approach is tested on a dummy synthetic data and a small materials data set of formation energies of transparent conducting oxides. The final goal is to identify the domain of applicability (DA) of three kernel-based ML models. As it's clearly stated, the challenge of DA in any ML research is timely and of interest. Thus, I believe the approach is relevant and arguably new. The text is also well written and it's easy to understand. It conveys a generally acceptable ML workflow for training and validation.

R3.2 We thank Reviewer 3 for their kind comments.

However, the discussions in the paper are mainly focused on the application of the DA approach as a fair metric to compare models and feature representations. I believe a general approach to DA is way more important than its application in comparing methods on its basis. There are a few recent research papers that try to address this issue from different aspects:

- Kailkhura et. al., "Reliable and Explainable Machine Learning Methods for Accelerated Material Discovery", Arxiv 2019;
- Haghightlari et. al., "Thinking Globally, Acting Locally: On the Issue of Training Set Imbalance and the Case for Local Machine Learning Models in Chemistry", ChemRxiv 2019.

R3.3 We thank the reviewer for pointing to the potential application of DA identification in the context of forming ensembles of local models as explored in the two provided references. Indeed, the proposed DA analysis could provide an advantage here, as it takes into account the information of model performance into the definition of sub-populations. We discuss this possibility now on Pg. 9 as potential direction for future work:

"A further potential application of the proposed approach is to form ensembles of different local models, each of which is only contributing to the overall predictions for data points inside its DA. The general approach of forming ensembles of local models is an emergent idea to cope with heterogeneous material classes [Kailkhura et al., 2019, Haghightlari et. al., 2019]. So far these efforts have focused on a priori² partitioning of the training set into sub-populations identified by automated clustering methods or prior knowledge, followed by fitting local models using the same regression technique for all sub-populations. In contrast, the DA approach can be used to incorporate the relative model advantages and disadvantages into the partitioning of the materials class."

Beyond this, we believe that the evaluation and comparison of models and material representations is an important problem in its own right as well as a necessary first step before the exploration of applications in model ensembles. This is based on our own experience as well as through extensive exchange with other colleagues working in representation design.

Since it's difficult to provide benchmark data for unsupervised techniques, these references can be used to evaluate the generalizability of the proposed approach.

²That is, the sub-populations are found solely based on the feature and property values and are not defined based on the relative performance difference between different representations and regressors.

R3.4 Regarding the evaluation, we stress again that DA identification procedure is not an unsupervised technique (see R3.1). Moreover, as noted by Reviewer 2, we use a standard process to evaluate the generalizability of the identified DAs: the DAs are evaluated using a separate hold-out dataset (referred to as DA validation set) that has not been involved in identifying the selector. Both of these sets are disjoint from the original model training sets, so that that the ML model losses involved in DA identification and evaluation are unbiased estimates of the true model performance.

Considering the novelty of the work, I recommend the following major concerns and comments be addressed before any potential publication: 1) It seems that each model is trained only once (on one of the training folds) and the SGD-DA approach will be applied to the errors of the model and splits of the test set. It is not clear that the prediction of the test set by ML model is required to validate the DA or not? If yes please also comment on the computational cost for each of the three feature representation methods.

R3.5 Each ML model is trained only once on the model training set via a standard procedure (e.g., internal cross-validation to optimize hyper-parameters). As usual, the (global) performance of the ML model is then evaluated on a hold-out test set, and for this step, the ML model representations have to be computed for the test points. At this point, the DA optimization and validation can be performed as a by-product from the labels and ML predictions of the test set. This involves, as mentioned in R3.4, a further split of the model test set into DA identification and DA validation set. Specifically for the results of the paper, this is the partitioning that is repeated multiple times via cross-validation to investigate the stability of the identified DAs. However, also with this repetition of the DA identification step, model representations do not have to be computed more often than in the usual ML workflow. This is now clarified on Pg. 3 of the main text:

“The DA optimization and validation can be performed as a by-product from the labels and ML predictions of the test set.”

2) The follow-up comment is on the optimization algorithm in the SGD method. I understand the objective function is explained here, but it seems that the core of the work (optimization algorithm) has been left out based upon the previously developed, SGD, technique. However, authors spend a generous amount of space to explain feature representations that are developed by other researchers.

R3.6 We thank the reviewer for this useful observation. The overview of the investigated representation is included to provide the necessary background for the discussion of the results. Pg. 3/4 of the revised manuscript now contains an overview paragraph regarding SGD optimization including a reference to additional technical details now provided in the SI:

“Optimizing the impact function over all conjunctive selectors that can be formed from a given set of base proposition is an NP-hard problem. This implies that there is no solver for it with worst-case polynomial time complexity (unless $P=NP$). However, there is a practically efficient branch-and-bound algorithm based on: 1) a non-redundant branching operator that generates exactly one selector for any set of selectors describing the same subset of data points, and 2) a tight bounding function that estimates an upper bound to the best attainable specialization of a given selector. This approach turns out to be very fast in practice—in particular, substantially faster than the model training process (see SI Section IV for more details).”

3) One of the main concerns is regarding the limited number of features that are used as DA selectors. Although authors justify the neutrality and interpretability of the features, there is a place of concern for the correlation between these features and actual features that have been used for the training. This can be discussed from three different angles: a) The choice of selectors may be biased to particular domain knowledge and perform in favor of one of the descriptors. This may be a valid point from a chemist point of view, but I believe it’s not really fair for the sake of comparison. One can choose selectors that are more correlated with their choice of feature representation.

R3.7 We agree with the reviewer that it is important to keep in mind that all findings are relative to the chosen DA representation, and other representations could reveal additional properties of the investigated material representations. We have further clarified our chosen DA representation in the main text, as discussed in R3.9. But we note here that generally, we are not suggesting that some representations are simply better than others, but instead observe qualitative differences between them as describable by the chosen DA language. Moreover, we would like to point out that including features that are highly correlated to a model representation (or even exact copies of the model features) does not necessarily lead to the identification of better DAs: consider for instance the example of predicting the area of rectangles just by one of their side-lengths (say a). An ordinary least squares model is then accurate only for rectangles around the expected value of the other side-length b (according to the modeled distribution). However, one cannot describe this domain of applicability in terms of the feature a .

b) Although the actual features may not be chemically interpretable, they are the same features that one needs to calculate before using the model to predict energies. Thus, it sounds Ok to identify DA based on the same features.

R3.8 While the ML representation features need to be computed in the overall process, they serve a different purpose than the DA features. In addition of being not readily interpretable, which contradicts one of our main design goals, they also substantially complicate the sampling of new points from the DA because the model descriptor/representation would have to be generated for each sample.

c) **The number of descriptors in each of the methods are not same. MBTR provides at least two times more descriptors than the other two (1400 vs 600). This is way more amount of information extracted to be interpreted in a small subgroup and against a limited number of selectors.**

R3.9 The reviewer brings up a good point that actually demonstrates the strength of the DA approach: the representation space inputted into subgroup discovery can be adapted for various purposes depending on the focus on the investigation. Because of we are interested in trying to detect differences in DAs for representations, we chose features corresponding to the unit cell structure (e.g., on the lattice vectors, lattice angles, and bond distances), which allows us to investigate how these representations specifically encode the geometries in different ways.

Regarding item #3, While exploring extended DA representations is certainly a useful direction of future research, one of our main design goals in the present work is the description of DAs in as simple as possible terms, which is intended to allow to easily interpret, compare, and to sample from them (e.g., for focused screening). This precludes the reviewer’s suggestion to replicate the model representation for the DA identification. We emphasize again that extending the DA representation is a worthwhile research direction, but meanwhile, the unit cell and compositional features are a good solution for our immediate design objectives. Motivated by the reviewer’s comment, we have added the following text to Pg. 4 of the manuscript:

“The description of DAs in as simple terms of the unit cell structure and composition allows for easily interpreting, comparing, and to sample from them (e.g., for focused screening). However we note that the representation space inputted into subgroup discovery can be adapted for various purposes depending on the focus on the investigation.”

The other main concern is about the interpretation of the results. As mentioned in the text, the coverage of the MBTR is almost half of the SOAP model/representation (%44 vs %76). Thus, the MBTR model is applicable in a subspace of the data, which is half of the size of coverage by SOAP. Thus the DA of the MBTR is too restricted. I think a higher coverage of the training data is more desirable for an ML model with acceptable performance.

R3.10 We agree that this is a reasonable requirement in some contexts whereas in others a higher error reduction might be more important. Therefore, we now point out on Pg. 3 of the revised manuscript that, as a simple modification of the SGD algorithm, a numerical trade-off parameter can be introduced to re-balance the relative importance of coverage and error reduction:

“With the above objective function we reduce the bi-criterial coverage/effect optimization problem to a uni-criterial impact optimization problem where both individual criteria are equally weighted and non- compensatory, i.e., due the multiplicative combination, very low values of one criterion cannot be compensated by very high values in the other. The relative weight of both criteria can be re-calibrated by introducing a simple exponential weight parameter (see SI Section III for a detailed discussion).”

As we discuss now in the SI, varying this parameter allows to find the DA Pareto front, i.e., the set of all DA selectors that are not dominated by another selector in terms of both, coverage and error reduction. Thus, the proposed method can be used to find the optimal DA for all possible preferences between the two criteria.

I suspect if we shrink the subspace of the SOAP model elaborately (e.g., from %76 to %50), the number of outliers and thus the subgroup error will also shrink.

R3.11 The reviewer’s suspicion is correct. In Fig. 3 of the SI, we now explore the attainable trade-off curve for all models based on the modification mentioned in R3.10. On the 44% coverage level, an equally good error reduction can be achieved for SOAP as for MBTR. However, in contrast to MBTR, SOAP allows an even better trade-off at the higher coverage level of 76% according to uniform criterion weighting (which is used in the main text).

In addition, there are a number of minor issues that should be addressed before the paper is accepted: 1) on page 6, line 2: impact needs to be italic?; 2) on page 9: Figure2, in the caption: the standard deviation based on the mathematical expression is ‘epsilon‘ not ‘s‘ 3) Please comment on the accessibility of the codes? or any GitHub repository for SGD/DA method?

R3.12 We thank the reviewer for pointing these typos out. We have addressed these issues in the revised version. All data and scripts involved in producing the results presented herein will be made available on github. An implementation of the SGD algorithm is publicly available through the realKD Java library at <https://bitbucket.org/realkd/realkd/> and an notebook illustrating the DA-identification procedure will be available at <https://analytics-toolkit.nomad-coe.eu>. These aspects are now mentioned in the ”Data and code availability” section of the main text.

REVIEWERS' COMMENTS:

Reviewer #2 (Remarks to the Author):

Thank you for the updates to the manuscript.

I am supportive of publication with minor changes noted below

* The statement "the relative error is guaranteed to be uncorrelated with the target values" would only be true under some additional about the origin of the errors. I do not see why this would be guaranteed for this data with these models. You should either justify this statement or soften it.

* I will note: it is still not clear to me whether relative or absolute error is preferred for finding a DA. However, with the addition of the figures in supplemental for DAs from absolute errors (Fig 4, Table III and IV), this is something the reader can investigate for themselves.

* Nit: The caption in Figure 4 refers to something that does not exist in the figure: "number of sample in the entire test set are also indicated"

Reviewer #3 (Remarks to the Author):

The revised version of the paper looks good. I have no extra comments on the paper.

NCOMMS-19-29036—Response to Remaining Referee Comments

Reviewer 2

The statement “the relative error is guaranteed to be uncorrelated with the target values” would only be true under some additional about the origin of the errors. I do not see why this would be guaranteed for this data with these models. You should either justify this statement or soften it.

As suggested by the reviewer the statement has been softened to:

Here, we focus on DA identification based on the relative error, as it is less correlated with the target values than the absolute error.

I will note: it is still not clear to me whether relative or absolute error is preferred for finding a DA. However, with the addition of the figures in supplemental for DAs from absolute errors (Fig 4, Table III and IV), this is something the reader can investigate for themselves.

We are happy that the referee agrees that the current solution appropriate given that a case for either variant can be made.

Nit: The caption in Figure 4 refers to something that does not exist in the figure: “number of sample in the entire test set are also indicated”

This caption of this figure has been corrected to:

The distributions of the three features referenced in the selector are shown with sub-population selected by condition in blue, sub-population deselected in gray, and threshold by red line.

Reviewer 3

The revised version of the paper looks good. I have no extra comments on the paper.

We are happy that the revised manuscript has satisfied the referee.